# Probiotics Enhance Bone Growth and Rescue BMP Inhibition: New Transgenic Zebrafish Lines to Study Bone Health

**DOI:** 10.3390/ijms23094748

**Published:** 2022-04-26

**Authors:** Jerry Maria Sojan, Ratish Raman, Marc Muller, Oliana Carnevali, Jörg Renn

**Affiliations:** 1Department of Life and Environmental Sciences, Università Politecnica delle Marche, Via Brecce Bianche, 60131 Ancona, Italy; j.m.sojan@pm.univpm.it (J.M.S.); o.carnevali@staff.univpm.it (O.C.); 2Laboratoire d’Organogenèse et Régénération, GIGA-I3, B34, Université de Liège, 4000 Liège, Belgium; ratish.raman@uliege.be (R.R.); joergrenn@yahoo.de (J.R.)

**Keywords:** *Danio rerio*, zebrafish, transgenic lines, bone matrix, probiotics, mineralization, BMP inhibitors, bone growth

## Abstract

Zebrafish larvae, especially gene-specific mutants and transgenic lines, are increasingly used to study vertebrate skeletal development and human pathologies such as osteoporosis, osteopetrosis and osteoarthritis. Probiotics have been recognized in recent years as a prophylactic treatment for various bone health issues in humans. Here, we present two new zebrafish transgenic lines containing the coding sequences for fluorescent proteins inserted into the endogenous genes for *sp7* and *col10a1a* with larvae displaying fluorescence in developing osteoblasts and the bone extracellular matrix (mineralized or non-mineralized), respectively. Furthermore, we use these transgenic lines to show that exposure to two different probiotics, *Bacillus subtilis* and *Lactococcus lactis*, leads to an increase in osteoblast formation and bone matrix growth and mineralization. Gene expression analysis revealed the effect of the probiotics, particularly *Bacillus subtilis*, in modulating several skeletal development genes, such as *runx2*, *sp7*, *spp1* and *col10a1a*, further supporting their ability to improve bone health. *Bacillus subtilis* was the more potent probiotic able to significantly reverse the inhibition of bone matrix formation when larvae were exposed to a BMP inhibitor (LDN212854).

## 1. Introduction

Probiotics are beneficial microbes that contribute health benefits to the host when provided in suitable quantities [1]. Bone growth and health are proven to be affected by probiotics since they rely on the gut mainly for the absorption of minerals and vitamins [2]. The novel term “osteomicrobiology” was coined for microbiota and bone health research [3]. There are many reports on the positive effects of various probiotic bacteria strains on bone health in various animal models and human studies [4,5,6,7,8]. Lactobacillus strains and Bifidobacterium were proven to prevent ovariectomized (OVX)-mediated bone loss in mice and rat models [9,10,11,12]. *Bacillus subtilis* supplementation was able to reduce bone loss due to periodontitis in rats [13]. In humans, there are multiple reports on the prevention of bone loss by various probiotics in post-menopausal women [14,15]. In zebrafish, Maradonna and others showed an increase in calcification after probiotic administration [16]. There are various possible modes of action for probiotics on bones. Osteoimmunology is a field of study that explores the close association between immune and skeletal systems and has found inflammatory conditions are associated with osteoporosis [17]. Some Lactobacillus strains have been shown to increase vitamin D receptor (VDR) expression by human and mouse epithelial cells [18]. Probiotics, through a possible interaction with oestrogens, inhibit bone loss linked to steroid deficiency, as was previously demonstrated in studies with mice [19]. The gut microbiome and its interaction with dietary calcium was also previously shown as another way to affect bone health, since calcium is essential for the maintenance of bone health by decreasing bone resorption [20]. Some beneficial strains are also an important source of vitamin K2, which acts as a cofactor in carboxylation of the matrix protein BGLAP (osteocalcin), thereby supporting bone mineralization [21,22,23].

The zebrafish (*Danio rerio*) has been increasingly used as a model species for skeletal development, since the basic regulatory networks and metabolic pathways are largely conserved between fish and mammals. A number of mutants have been described in zebrafish that mimic human pathologies such as osteoporosis, osteopetrosis, osteoarthritis [24], thus illustrating how homologous genes play similar roles in both species. In addition, zebrafish larvae can advantageously replace cell culture to test pro- or anti-osteogenic properties of specific compounds because they can reproduce the complex regulatory interactions taking place between different tissues. Although developing larvae may be fixed at different stages to undergo specific staining for various tissues and features (cartilage, bone matrix, etc.), the optical clarity of zebrafish embryos and larvae allow for continuous observation of live animals. To that purpose, several transgenic lines are available that express a fluorescent protein (GFP, mCherry, citrine) under the control of a synthetic or natural transcription regulatory region [25,26]. Specific transgenic lines have revealed in vivo activation of canonical BMP, Hedgehog, or Wnt pathways [27,28,29,30,31] during bone development. Others have used cell-specific promoter regions or recombinant bacterial artificial chromosomes (BACs) to target the expression of the reporter protein to chondrocytes, early or late osteoblasts, or osteoclasts [32].

Even with all the reported therapeutic effects of probiotics or gut microbiota on bones, there is still a lack of a clear understanding on how they are able to influence bone homeostasis [33]. Here, we present two newly generated transgenic zebrafish lines that were obtained by inserting a reporter protein coding sequence directly into the coding region of two endogenous genes. These new transgenic lines express the GFP protein under transcriptional control of the endogenous regulatory regions for (i) the osteoblast marker *sp7* (Sp7 transcription factor) gene and (ii) its downstream target gene *col10a1a* encoding the osteoblast- and hypertrophic chondrocyte-specific collagen type X alpha 1a chain. We describe the expression pattern of each reporter gene and apply them to testing the efficacy of osteogenic strains of probiotics. We analysed the modulatory effects of two probiotics, *Bacillus subtilis* and *Lactococcus lactis*, on osteoblast differentiation and early skeletal growth in zebrafish using these two lines. Furthermore, we tested the ability of each probiotic to counteract the deleterious effect of BMP inhibitor treatment on the bone matrix. The results presented here emphasize yet again that zebrafish, particularly transgenic lines, are an ideal model for live studies on skeletogenesis, including the impact of probiotics.

## 2. Results

### 2.1. Generation and Characterization of New Transgenic Lines

#### 2.1.1. *Tg(col10a1a:col10a1a-GFP)* Line

We generated the *Tg(col10a1a:col10a1a-GFP)* transgenic line where the expression of a fusion protein between Col10a1a and GFP is driven by the endogenous zebrafish *col10a1a* promoter (Figure 1a,b). To characterize this line, we crossed it with the *Tg(Ola.Sp7:mCherry)* line that expresses the red fluorescent mCherry protein in osteoblast cells [34]. We found GFP expression localized very specifically in the same regions as osteoblasts expressing the mCherry protein. This includes the early appearance of the cleithrum, opercle and pharyngeal tooth bud at 3 days post-fertilization (dpf), the parasphenoid and branchiostegal rays at 4 dpf, as well as maxillary, dentary, and entopterygoids at 5–6 dpf (Figure 1c), consistent with previous in situ hybridization studies [35]. Alizarin red (AR) live fluorescent staining of *Tg(col10a1a:col10a1a-GFP)* larvae (Figure 1d) confirmed the presence of GFP in mineralized bone matrix.

The Col10a1a-GFP protein produced by the *Tg(col10a1a:col10a1a-GFP)* line contained the original Col10a1a N-terminal signal peptide that would lead to its secretion into the extracellular space, which suggested that it would be secreted from the producing cells and bound by the nearby extracellular bone matrix to generate the observed labelling pattern. To investigate this hypothesis in detail, we turned back to the *Tg(col10a1a:col10a1a-GFP*; *Ola.Sp7:mCherry*) double transgenic line and dissected individual bone elements for analysis. Looking at the developing cleithrum (cl) (Figure 1e), we observed that most of the Col10a1a-GFP protein was associated with the extracellular bone matrix. This conclusion was even more apparent when looking at the developing opercle (Figure 1f). Here, the Col10a1a-GFP was mainly present in the proximal part that contained less cells, whereas the majority of mCherry-expressing osteoblasts were found at the distal growth fringe of the opercle. In only some cases, fluorescence could be observed inside cells. This predominant bone matrix staining was further confirmed by comparing the Col10a1a-GFP pattern in *Tg(col10a1a:col10a1a-GFP)* transgenic larvae with AR staining performed on the same animal (Figure 1g,h).

In the next step, we wondered whether the fact that the transgene was expressed in osteoblasts would favour its preferential binding to the nearby bone matrix. Therefore, we engineered a synthetic mRNA coding for the Col10a1a-GFP fusion protein that we directly injected into fertilized eggs so that it would be translated in every cell within the embryo. Microinjection of a control mRNA coding for GFP lead to an intense, widely distributed green fluorescence at 5 dpf (Figure 1i). In contrast, microinjection of the col10a1a-GFP mRNA resulted in weak fluorescence specifically restricted to the cleithrum and the opercle (Figure 1i, right), further supporting the notion that this fusion protein specifically binds to the bone matrix. Finally, to test the binding of Col10a1a-GFP to unmineralized bone matrix, we took advantage of the finding that the *entpd5* gene is required for bone mineralization [36]. We designed morpholino antisense oligonucleotides against *entpd5* mRNA and injected them into fertilized eggs derived from a heterozygous *Tg(col10a1a:col10a1a-GFP)* parent. As expected, about half of the larvae (44/85) revealed Col10a1a-GFP fluorescence in control-injected larvae, while 50/109 displayed similar fluorescence in *entpd5* morphants (Figure 1j, top), indicating that *entpd5* knockdown did not affect bone staining. In contrast, AR staining for mineralized bone was completely absent or very weak in all morphants when compared to the control-injected larvae (Figure 1j, bottom).

#### 2.1.2. *Tg(sp7:sp7-GFP)* Line

Using the same CRISPR/Cas9 method, we generated another transgenic line by targeting the insertion of GFP reporter cDNA into the endogenous *sp7* coding region, resulting in a line expressing a fusion protein between Sp7 and GFP (Figure 2a). This new line, *Tg(sp7:sp7-GFP)*, was analysed for green fluorescence in parallel with red fluorescence in the *Tg(Ola.Sp7:mCherry)* line. Comparison of the two lines (Figure 2b) revealed that the expression of both transgenes largely overlapped, starting at 3 dpf in the cleithrum and opercle and extending to the maxillary, dentary, branchiostegal rays and entopterygoids at later stages. However, some differences in the expression pattern were also apparent, mainly earlier expression in the pharyngeal tooth bud at 3 dpf and stronger expression in the entopterygoid at 6 dpf in the *Tg(sp7:sp7-GFP)* line. Both lines displayed weakened expression at 10 dpf and beyond (not shown).

### 2.2. Effect of Probiotics

#### 2.2.1. Effect of Probiotics on Bone Formation

In a preliminary experiment, we used WT zebrafish larvae to test the effect of probiotics on specific mRNA levels. Total RNA was extracted from 7 dpf larvae grown in control (E3 + water) or E3 supplemented with *Bacillus subtilis* (BS) or *Lactococcus lactis* (LL) probiotics. Using RT-PCR, we observed that all bone-related gene mRNA levels were significantly increased upon exposure to the probiotics. Interestingly, expression of the specific marker genes *sp7*, *col10a1a*, *spp1* and *runx2b* were more extensively induced by BS, but *bglap* mRNA levels were not significantly affected. In contrast, induction of *cyp26b1*, coding for an enzyme degrading retinoic acid, was significantly higher with LL (Figure 3).

To facilitate and complement this observation, we decided to test the effects of the two different probiotics on bone development by direct live observation of developing bone elements using three transgenic lines. Two lines were based on the osteoblast-specific *sp7* promoter, one from medaka, *Tg(Ola.Sp7:mCherry)* [34], and the other from the endogenous zebrafish *sp7* gene, *Tg(sp7:sp7-GFP)* presented above. The third line was the *Tg(col10a1a:col10a1a-GFP)* line presented above, based on the endogenous *col10a1a* promoter, which preferentially reveals unmineralized or mineralized bone matrix.

Control and probiotic enrichment conditions were applied to individuals of each of the three transgenic lines. For the two transgenic lines based on the *sp7* promoter, *Tg(Ola.Sp7:mcherry)* and *Tg(sp7:sp7-GFP)*, we determined the integrated pixel intensity in the head areas (lateral and ventral view) at 7 dpf (Figure 4a,b). We observed a significant increase in fluorescence upon exposure to both BS and LL probiotics in the head and opercle areas in lateral views, but areas in ventral views achieved significance only in the *Tg(Ola.Sp7:mCherry)* line. In all cases, the increase was consistently more intense with BS than LL. Representative images of larvae from the corresponding conditions illustrate the measured trends (Figure 4c). Using the *Tg(col10a1a:col10a1a-GFP)* line, we decided to extend the observations to a later stage, and 10 dpf was selected since most cranial bone elements are detectable. In addition, we performed live AR staining before observation in order to further illustrate predominant staining of the bone matrix in this line. The pixel intensities were significantly higher than the control in all areas for both probiotics (Figure 4d). Furthermore, BS caused a significantly higher GFP pixel intensity compared to LL in the ventral view. Simultaneous staining with AR confirmed that mineralization in the BS-treated larvae had the highest integrated pixel intensity in the total head (lateral and ventral). Representative images of larvae from the corresponding conditions illustrate the measured trends (Figure 4e), while merged images confirm the near-perfect overlap of GFP and AR fluorescence for both signal positive areas as well as pixel intensity.

#### 2.2.2. BMP Inhibitor Exposure Followed by Probiotic Treatment

BMP signalling is known to be required for osteoblast differentiation and for bone mineralization [37,38]. The most eminent marker gene widely used for investigating osteoblast differentiation in mammals and teleost species is the *sp7* gene [39,40,41]. We treated *Tg(Ola.Sp7:mCherry)* transgenic larvae with the BMP inhibitor LDN212854 from 2 dpf until 4 dpf at two different concentrations (10 µM and 20 µM) to test its effect on the osteoblast population. We observed a weak but significant increase in mCherry expression in the 20 µM concentration group when compared to the control (DMSO) group at 5 dpf (Figure 5), suggesting that, at this concentration, osteoblast proliferation and/or differentiation is affected by BMP inhibition.

Furthermore, we used the *Tg(col10a1a:col10a1a-GFP)* line to observe the effects of the BMP inhibitor LDN212854 at 20 µM on bone matrix formation and mineralization, and to investigate the potential protective properties of the probiotics. LDN212854 was administered from 2 dpf to 4 dpf, followed by probiotic supplementation (BS or LL) from 5 dpf to 10 dpf. DMSO was used as a second control since the inhibitor was dissolved in DMSO. Sampling was performed at 10 dpf, and additional staining with AR was used to visualise mineralized structures in the larvae. Compared to both control and DMSO, we observed a dramatic decrease in the integrated pixel intensity values in all analysed areas in the presence of LDN, both for Col10a1a-GFP and live AR staining. Compared to BMP inhibition alone, additional treatment with probiotics resulted in significantly increased *Col10a1a-GFP* fluorescence in both lateral and ventral observation, but the increase in live AR staining never reached significance. These observations suggest that probiotics, particularly BS, can revert the deleterious effect of BMP inhibition on bone matrix formation, while AR staining indicated that the effect was not evident for mineralization (Figure 6a,b).

## 3. Discussion

Transgenic zebrafish reporter lines with specific expression of fluorescent proteins for following the development of specific organs, tissues, and cells in living larvae are now widely used in developmental biology. Here we present two new transgenic lines, namely, *Tg(sp7:sp7-GFP)* and *Tg(col10a1a:col10a1a-GFP)*, which we obtained by inserting the GFP coding sequence into the endogenous *sp7* or *col10a1a* gene coding sequence with CRISPR/Cas9 technology. We used these lines to visualize formation of the zebrafish skeleton and to evaluate the effects of two types of probiotics on this process, first in the absence and then in the presence of a BMP inhibitor.

Several transgenic zebrafish lines expressing fluorescent reporter genes under the control of the sp7 promoter have been described in recent years. However, there are important differences between these lines and the *Tg(sp7:sp7-GFP)* line presented here: (i) previously described lines, such as the *Tg(Ola.Sp7:mCherry)* [34,42] or *Tg(Ola.Sp7:mCherry-NTR)* [43] use the heterologous promoter from medaka (*Oryzias latipes*, Ola) to drive expression of the transgene [44]; (ii) these lines were obtained by insertion of an artificial construct at one or more random, unknown locations in the genome. In contrast, the *Tg(sp7:sp7-GFP)* carries the reporter gene in place of the endogenous *sp7* gene, it is under the control of the endogenous regulatory regions, and thus, should reproduce more correctly the *sp7* expression pattern. Indeed, consistent with the pattern observed here in the transgenic line, in situ hybridization has revealed *sp7* expression in the tooth buds at 4 dpf [45], and in the maxillary [46,47] and the entopterygoid at 3 dpf [39]. In comparison, the higher mCherry expression observed in earlier studies could be due to the random location of the transgene. Despite these differences in the exact timing of expression patterns, these two lines display fluorescence in many overlapping regions, such as the opercle, cleithrum or branchiostegal rays [46,47].

Col10a1a is a secreted protein, and in our *Tg(col10a1a:col10a1a-GFP)* zebrafish, GFP insertion preserves the N-terminal signal peptide of Col10a1a (see Figure 1a,b). Thus, these transgenic fish encode a fusion protein that is secreted by cells. Although we observed some fluorescent cells in this line, it appears that the main fluorescent structure is the extracellular bone matrix. The Col10a1a-GFP fusion protein strongly binds to the extracellular bone matrix after secretion, as shown by an overlapping pattern with AR staining, irrespective of where it was expressed, since microinjection of *col10a1a-GFP* mRNA led to expression in essentially all embryonic cells and resulted in the same specific fluorescent staining of bone elements. We further showed with knockdown of the *entpd5* gene [36] that bone mineralization was not required for fluorescent labelling of the bone matrix (Figure 1j). Therefore, we consider the *Tg(col10a1a:col10a1a-GFP)* line as the first reporter line that reveals the extracellular bone matrix, mineralized or not. As such, it represents an important tool for analysing the sequential events of osteoblast differentiation, bone matrix deposition and its mineralization in live larvae when combined with transgenic lines such as the *Tg(sp7:sp7-GFP)* and AR staining.

A preliminary test for the effects of probiotics revealed that the transcription of several marker genes for bone development were significantly upregulated upon probiotic treatment at 7 dpf. Interestingly, expression of the pro-osteogenic genes *sp7*, *col10a1a*, *spp1* and *runx2b* was significantly higher after BS treatment, but *cyp26b1* expression was significantly higher after LL treatment. The *cyp26b1* gene codes for a protein with retinoic acid 4 hydroxylase activity, whose mutation leads to severe defects in head cartilage formation [48], increased ossification of the vertebral column [42,49] with complex effects on skeletal formation, depending on timing and location [50]. It is therefore difficult to predict how this differential response to the different probiotics will affect skeletal development. None of the probiotics affected *bglap* expression by mature osteoblasts compared to a more predominant effect on *spp1* expressing immature osteoblasts; this could be due to the degree of osteoblast differentiation at the particular stage studied here [51]. Since *runx2b* expression has to be downregulated for immature osteoblasts to differentiate into mature osteoblasts and formation of mature bone [52], the stage of osteoblast differentiation observed here appears to be more transitional, from immature to mature, with high upregulation of *spp1* and *sp7* and no downregulation of *runx2b* by both probiotic treatments that was more significant for BS. All this preliminary evidence from the expression pattern of genes related to skeletal development directed us to further explore the possibility of using reporter lines to confirm the results.

Transgenic reporter lines offer the opportunity to follow the formation of specific tissues in living embryos and larvae over time and in specific locations. The two new lines presented here both drive their transgene expression from endogenous regulatory regions and reveal either the location of osteoblasts (*Tg(sp7:sp7-GFP)* line) or of the bone matrix (*Tg(col10a1a:col10a1a-GFP)* line). When we tested probiotics exposure with these transgenic lines, BS was found to significantly induce *sp7*-driven expression in osteoblasts at 7 dpf and Col10a1a-GFP labelling of the bone matrix at 10 dpf. These results observed for the bony structures of the head are clearly in agreement with the mRNA results. Additional AR staining of the *Tg(col10a1a:col10a1a-GFP)* line revealed that probiotics induced a significant increase in mineralized bone matrix as well. The positive influence of BS on bone matrix formation and mineralization was significant when the bony structures were analysed from both lateral and ventral views of the head, whereas the weaker effect of LL produced non-significant effects in both views. This indicates that the different bacteria have varying abilities for modulating the process of bone formation, and in our study, *B. subtilis* was more efficient at positively influencing osteogenesis than *L. lactis*. In addition, we showed that treatment with the BMP inhibitor LDN212854 dramatically decreased bone matrix labelling by Col10a1a-GFP and bone mineralization as assessed by AR. In contrast, BMP inhibitor treatment of the *Tg(Ola.Sp7:mCherry)* line had little or no effect on osteoblast-specific expression (Figure 5), in line with previous observations showing that the BMP inhibitors dorsomorphin and K02288 decreased bone mineralization without affecting osteoblast numbers [38]. Although BMP signalling is required for a vast number of events in early embryogenesis, treatment from 2 dpf does not affect the general morphology, as illustrated by the lack of an effect on cranial cartilage formation as previously observed [38]. Studies in mice, using various conditional gene knock outs or transgenic expression of BMP inhibitors, have also indicated that BMPs may play dual roles, facilitating osteoblast differentiation by inducing Runx2 and Sp7 expression, and independently, osteoblast function by stimulating bone matrix production [53]. The finding that probiotic treatment induced both osteoblast regulators (*runx2b* and *sp7*) and bone matrix protein genes (*spp1* and *col10a1a*) hints at a possible mechanism for rescuing the negative effect of BMP inhibition on bone. Interestingly, we observed that BS was able to partially revert this effect on bone matrix deposition but could not rescue bone mineralization. These results suggest a decoupling of bone matrix deposition from bone mineralization that may be differentially affected by probiotics (or BMP signalling), possibly related to the absence of induction of the late marker *bglap* by probiotics.

Osteoblast or bone matrix reporter transgenic lines combined with staining techniques like AR are useful for following osteoblast differentiation, bone matrix deposition and bone mineralization simultaneously. Future studies may take further advantage of these transgenic lines by focusing on continuous monitoring of transgene expression during development and in adults or on specific bone structures. In this respect, studies in zebrafish can also provide important insights into skeletal development for mammals and humans; indeed, many examples of zebrafish genes playing similar roles to their human homologs have been described [24,25]. It is important, however, to note that a whole genome duplication occurred in teleosts, leaving many duplicated genes in fish genomes relative to mammals that may have functionally diverged. This situation requires either mutagenesis of both orthologs in zebrafish or, at least, a thorough analysis of their spatio-temporal expression pattern. In our study, the new transgenic lines *Tg(sp7:sp7-GFP*) and *Tg(col10a1a:col10a1a-GFP*) clearly evidenced the pro-osteogenic effects of two probiotics strains that were in agreement with the gene expression results. Thus, we can conclude that the probiotics are clearly pro-osteogenic, both alone and in the presence of a BMP inhibitor, with a clear advantage for BS (*Bacillus subtilis*). These findings open a new outlook for the use of probiotics as a prophylactic treatment for improving bone growth and health, which is currently a very under-explored area of research.

## 4. Materials and Methods

### 4.1. Generation of Transgenic Lines Using the CRISPR/Cas9 Method

To generate fluorescent reporter lines where the expression of the fluorescent protein GFP would be driven by endogenous bone-specific promoters, we engineered a plasmid containing the coding sequence for GFP and a specific sequence (Mbait) for which we also engineered a corresponding gRNA (gRNA1, see Figure 1a) [54]. By co-injecting the bait gRNA1, specific col10a1a or sp7 gRNA, plasmid, and *Cas9* nuclease into fertilized eggs as previously described [54], we generated double-stranded breaks within the endogenous gene and we linearized the plasmid with GFP cDNA. Injected individuals were then screened for fluorescence in bone structures, indicating that the GFP cDNA was inserted in frame and in the correct orientation within the endogenous target gene (Figure 1a). Positive individuals were grown, tested for germ line transmission into the F1 generation, and the exact sequence at the insertion point was determined. In each case, insertion of the GFP coding sequence resulted in a sequence coding for a fusion protein containing the N-terminus of the target gene and GFP expressed under the control of the *col10a1a* or the *sp7* regulatory regions (Figure 1a,b and Figure 2a).

Sequence of guide RNAs:Mbait: gRNA1: GGCTGCTGCGGTTCCAGAGG*col10a1a*: gRNA2: GGAGTAAGGCTGGTACTGCG*sp7*: gRNA3: GGCTCATTCAGCTCAAGCGG

Sequence of primers for insertion site sequencing:GFP-rev: GGTCTTGTAGTTGCCGTCGT*col10a1a*-for: TTGTCAAGAAGGTGATGAAGG*sp7*-for: AAAAGGCCTACAGCATGACTTC

### 4.2. Morpholino Injection

One to two cell-stage embryos were injected as previously described [55] with 3 ng of antisense morpholino oligonucleotides (MO, Gene Tools Inc., Philomath, OR, USA) complementary to the translational start site of the *entpd5* gene. Morpholinos were diluted in Danieau buffer and tetramethylrhodamine dextran (Invitrogen, Merelbeke, Belgium) was added at 0.5% to verify proper injection of the embryos by fluorescence stereomicroscopy. Standard control morpholino (MOcon) was injected at the same concentration. Although no increase of cell death was observed in the morphants, parallel co-injection experiments with 4.5 ng of a morpholino directed against p53 [56] were performed to ensure inhibition of MO-induced non-specific cell death [57]. The effects of morpholino injection were tested on at least 100 individuals performed in at least three independent experiments.

Sequence of the morpholino oligonucleotides:MO*entpd5*: AATTTAGTCTTACCTTTTCAGGCMOcon: random sequenceMOp53: GACCTCCTCTCCACTAAACTACGAT

### 4.3. RNA Extraction and Quantification

Total RNA was extracted from larvae (*n* = 8) using RNAeasy Microkit (Qiagen, Hilden, Germany) and eluted in 20 µL of molecular grade nuclease free water. Final RNA concentrations were determined using a nanophotometer (Implen, Munich, Germany). Total RNA was treated with DNase according to the manufacturer’s instructions (Sigma-Aldrich, St. Louis, MO, USA). One milligram of total RNA was used for cDNA synthesis using iScript cDNA Synthesis Kit (Bio-Rad, Hercules, CA, USA) and stored at −20 °C until further use as described previously [58].

### 4.4. Real Time PCR (RT-PCR)

RT-PCR reactions were performed with the SYBR green method in a CFX thermal cycler (Bio-Rad, Italy) in triplicate as described previously [59]. Primers were used at a final concentration of 10 pmol/mL. The thermal profile for all reactions commenced with 3 min at 95 °C, followed by 45 cycles of 20 s at 95 °C, then 20 s at 60 °C and 20 s at 72 °C. Dissociation curve analysis revealed a single peak in all cases. Ribosomal protein L13a (*rpl13a*) and ribosomal protein, large, P0 (*rplp0*) were used as the housekeeping genes to standardize the results by eliminating variation in mRNA and cDNA quantity and quality. No amplification product was observed in negative controls and primer-dimer formation was never seen. Data was analysed using iQ5 Optical System version 2.1 (Bio-Rad) including Genex Macro iQ5 Conversion and Genex Macro iQ5 files. Modification of gene expression between the experimental groups is reported as relative mRNA abundance (arbitrary units). All primer sequences used in the study are listed in Table 1.

### 4.5. Zebrafish Transgenic Lines Maintenance

Broodstock from the transgenic lines used in our experiments, *Tg(col10a1a:col10a1a-GFP*), *Tg(sp7:sp7-GFP)* and *Tg(Ola.Sp7:mCherry)*, were maintained at the zebrafish facilities, GIGA-R, University of Liège, Belgium in a recirculating water system (Tecniplast, Buguggiate, VA, Italy). To collect fertilised eggs in the morning, brooders were maintained at a 1:2 male to female ratio the day before and set for overnight breeding in tanks with slopes. Collected eggs were maintained in small tanks until hatching at 3 dpf. Using a fluorescent stereomicroscope (Olympus SZX10), larvae expressing the reporter proteins were screened and randomly distributed into 6 well plates at a density of 15 larvae per well with 10 mL of E3 medium per well until 7 dpf and then in small tanks with 15 larvae per 45 mL of E3 medium until 10 dpf. Seventy percent of the medium was exchanged daily. Commercial feed and live feed (paramecia) were administered from 5 dpf to 10 dpf along with the treatments.

### 4.6. Exposure to LDN212854 and Probiotics

Two probiotics, *Bacillus subtilis* (BS) and *Lactococcus lactis* (LL), were obtained from Fermedics (Machelen, Belgium) as lyophilized powder at a commercially formulated concentration of 10^11^ CFU/g. After preliminary tests, larvae were exposed at a concentration of 10^6^ CFU/mL administered in water from 5 dpf to 10 dpf. The type 1 BMP receptor inhibitor LDN212854 (Cat. No. 6151; Bio-Techne Ltd. TOCRIS, Bristol, United Kingdom) was dissolved in DMSO and used at concentrations of 10 µM and 20 µM, starting at 2 dpf until 4 dpf. Combined treatments were performed by exposing the larvae to LDN212854 from 2 dpf until 5 dpf, followed by probiotic treatment until 10 dpf. Each experiment was performed in triplicate.

### 4.7. Alizarin Red (AR) Staining

AR staining is one of the most common techniques for observing the extent of bone mineralization [60]. Larvae were sacrificed by exposure to MS-222 (ethyl 3-aminobenzoate methane sulfonate; Merck, Overijse, Belgium) and stained with 0.01% AR-S (Merck, Overijse, Belgium) for 15 min. They were then placed lateral side down onto glycerol (100%) for imaging.

### 4.8. Image Acquisition and Analysis

Imaging was performed with a Leica fluorescence stereomicroscope (Leica, Wetzlar, Germany) equipped with either a red fluorescence filter (λex = 546/10 nm, ET-DSR) for *Tg(Ola.Sp7:mCherry*) and AR-stained fish or a green fluorescence filter (λex = 470/40 nm) for *Tg(sp7:sp7-GFP)* and *Tg(col10a1a:col10a1a-GFP).* All images were acquired with a DFC7000T colour camera (Leica, Wetzlar, Germany) according to the following parameters: 24-bit coloured image, exposure time 2 s (green filter EGFP) or 1 s (red filter ET-DSR), gamma 1.00, image format 1920 × 1440 pixels, binning 1 × 1. Images were acquired using constant parameters and analysed using ImageJ version 2.1.0/1.53c software after splitting the colour channels of the RGB images. The green or red channel 8-bit images were adjusted uniformly for optimum contrast and brightness for improved visibility of the structures. The integrated pixel intensity was measured inside the total bone areas (in lateral and ventral view) of each fish and the integrated pixel intensity from the eye was subtracted. The values were further corrected for the head area (pixel intensity/head area) to eliminate possible inter-specimen size variability due to non-homogenous growth. The corrected values were plotted relative to the control arbitrarily set to 100%. A representative image showing how the pixel intensity was measured in both lateral and ventral head views is presented (Appendix A).

### 4.9. Statistical Analysis

Data from all groups were normally distributed, as assessed by a Shapiro–Wilk’s test (*p* > 0.05), and the variances were homogenous, as assessed by Levene’s test for equality of variances (*p* > 0.05). The differences between the control and the treatments were tested with a one-way analysis of variance (ANOVA) followed by Tukey’s post hoc test (*p* < 0.05) for all image analysis data and gene expression differences between groups. All the tests were performed using R version 4.0.2 and plots were generated using ggplot2 within R [61]. In our graphs, we use an alphabetic code to indicate statistically significant differences between groups in multiple comparison tests as a simple way to present all pair-wise comparisons. Every group is compared to every other group, and the groups that share the same letters are not statistically different. For instance, three treatments that are all significantly different from one another would be labelled a, b, and c; if two of the treatments differed from each other, but neither differed significantly from the third, they would be labelled a, b, and ab. The significance level was set at a constant *p* < 0.05 in all cases, but the focus was on all possible pair-wise comparisons.

## 5. Conclusions

We present two transgenic zebrafish lines based on insertion of a fluorescent protein coding cDNA into the coding regions of two bone-specific genes, *sp7* and *col10a1a*, which allow osteoblast formation and bone matrix growth, respectively, to be monitored in living animals. Using these two transgenic lines, the bone protection properties of two probiotics, *B.subtilis* and *L.lactis*, were revealed in addition to the specific ability of *B.subtilis* to counter the action of a BMP inhibitor. Our study therefore confirms the relevance of probiotics in promoting bone growth and bone health maintenance.

## Figures and Tables

**Figure 1 ijms-23-04748-f001:**
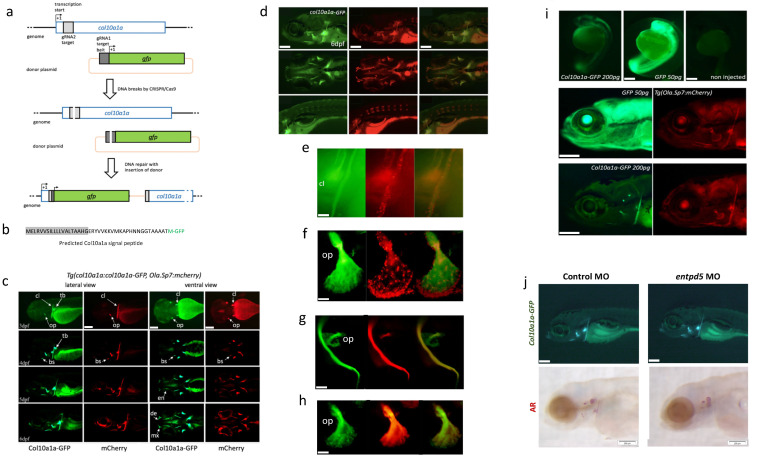
Characterisation of the *Tg(col10a1a:col10a1a-GFP)* transgenic line. (**a**) Schematic representation of the endogenous *col10a1a* gene (top line), the plasmids used for microinjection along with the two gRNAs (bait and *col10a1a*, respectively gRNA1 and gRNA2), the resulting cuts in the genomic DNA and plasmids, and the expected reporter construct in the transgenic genome. (**b**) N-terminal end of the fusion protein produced by the *Tg(col10a1a:col10a1-GFP)* transgenic line, with the predicted signal peptide in grey and the original GFP translational start site (M) in green. (**c**) Timeline of expression of GFP (green fluorescence) and mCherry (red fluorescence) in double-transgenic larvae *Tg(col10a1a:col10a1a-GFP*; *Ola.Sp7:mCherry)*. Lateral and ventral views at different developmental stages as indicated, anterior to the left. White arrows point to specific elements: (bs) branchiostegal ray, (cl) cleithrum, (de) dentary, (en) entopterygoid, (mx) maxillary, (op) opercle and (tb) tooth bud. (**d**) Expression of Col10a1a-GFP protein (green) in 6 dpf *Tg(col10a1a:col10a1a-GFP)* larvae live-stained with AR to visualize mineralized bone. (**e**–**h**) Close inspection of Col10a1a-GFP localization (green) compared to mCherry expression by osteoblasts in (**e**) the cleithrum (cl) or the opercle (**f**) of 9 dpf *Tg(col10a1a:col10a1a-GFP*; *Ola.Sp7:mCherry*) zebrafish larvae. (**g,h**) Expression of GFP protein (green) in *Tg(col10a1a:col10a1a-GFP*) 6 dpf larvae live-stained with AR to visualize bone matrix. Close inspection of the cleithrum (**g**) and the opercle (**h**). (**i**) Zebrafish *Tg(Ola.Sp7:mCherry)* larvae after microinjection of mRNA coding for GFP or for the fusion protein Col10a1a-GFP. Top: embryos at 1 dpf, showing weak fluorescence of Col10a1a-GFP, extremely strong fluorescence of GFP, and no fluorescence in controls. Bottom: the same larvae at 5 dpf, still showing strong GFP expression in the entire body and weak, but specific fluorescence of Col10a1a-GFP located in bone elements (cleithrum and opercle) as confirmed by the red fluorescence of osteoblast-specific mCherry. (**j**) Morpholino injection into *Tg(col10a1a:col10a1a-GFP*) larvae. The Col10a1a-GFP protein labels cranial bone elements at 4 dpf in both control and *entpd5* MO injected larvae (top), but AR staining is absent in *entpd5* morphants (bottom). The scale bars, given in the left bottom corner of the images, represent 200 µm, except for (**e**–**h**), where they indicate 20 µm.

**Figure 2 ijms-23-04748-f002:**
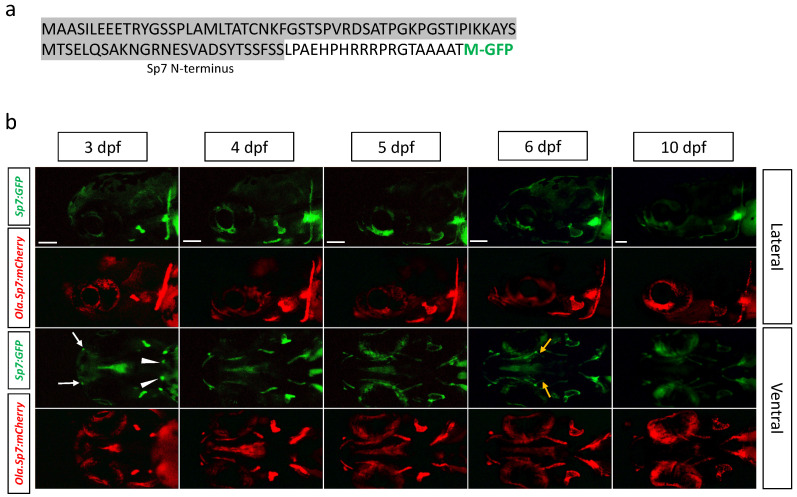
GFP expression in the *Tg(sp7:sp7-GFP)* transgenic line. (**a**) N-terminal end of the fusion protein produced by the *Tg(sp7:sp7-GFP)* transgenic line, with the predicted signal peptide in grey and the original GFP translational start site (M) in green. (**b**) Transgene expression in the *Tg(sp7:sp7-GFP)* and the previously described *Tg(Ola.Sp7:mCherry)* lines tracked in the head region (top: lateral view and bottom: ventral view) from 3 dpf to 10 dpf. Earlier transgene expression occurs in the maxillary (white arrows) and pharyngeal tooth buds (white arrowheads) at 3 dpf and in the entopterygoid (yellow arrows) at 6 dpf of the *Tg(sp7:sp7-GFP)* line. The scale bars, given in the left bottom corner of the images in the top row, correspond to 100 µm.

**Figure 3 ijms-23-04748-f003:**
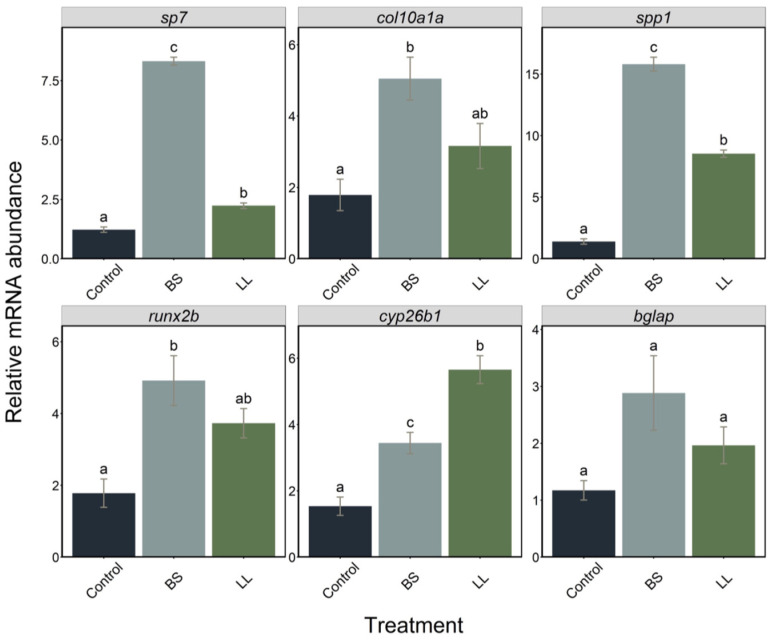
Relative expression levels of *sp7*, *col10a1a*, *spp1*, *runx2b*, *cyp26b1* and *bglap* genes in larvae (*n* = 7) treated with two probiotics, BS and LL, sampled at 7 dpf. Data are presented as mean ± S.D. One-way ANOVA and Tukey’s multiple comparison tests are used. Different letters denote statistically significant differences (*p* < 0.05) between the experimental groups.

**Figure 4 ijms-23-04748-f004:**
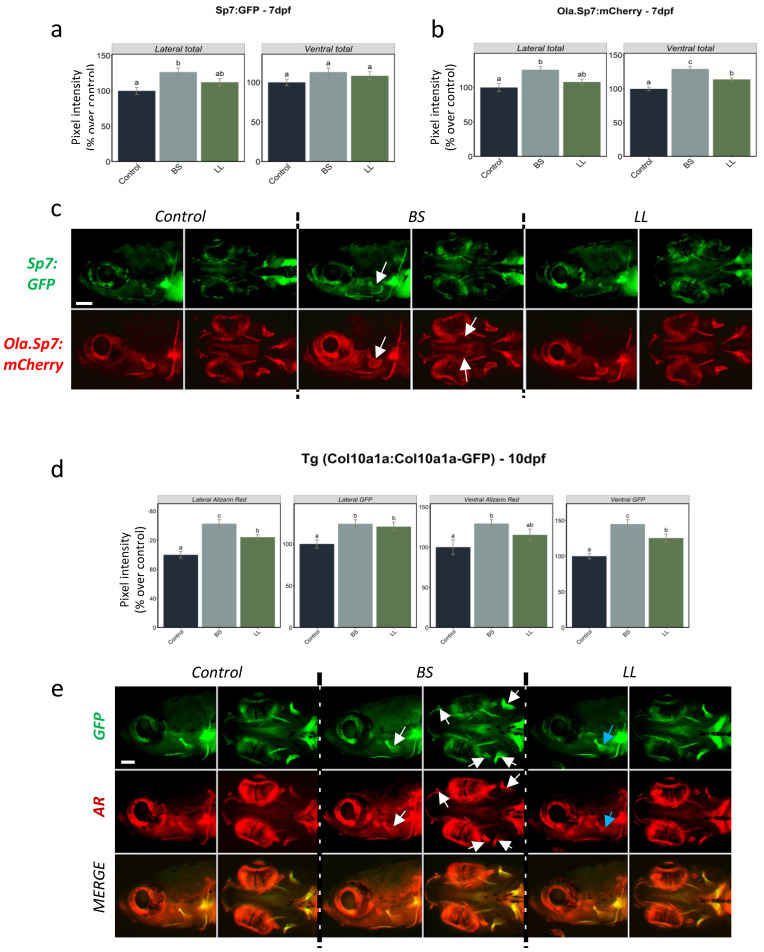
Integrated pixel intensity values for various areas measured in 7 dpf zebrafish from (**a**) *Tg(sp7:sp7-GFP)* and (**b**) *Tg(Ola.Sp7:mCherry)* larvae in controls and upon two different probiotic treatments. (**c**) Signal expression images (lateral and ventral views) of the head area of *Tg(sp7:sp7-GFP)* and *Tg(Ola.Sp7:mCherry)* larvae under the different conditions. (**d**) Integrated pixel intensity values for various areas measured in 10 dpf *Tg(col10a1a:col10a1a-GFP)* zebrafish larvae from the different treatment groups and stained with AR. (**e**) GFP, AR fluorescence and merged images of the head area (lateral and ventral views) of *Tg(col10a1a:col10a1a-GFP)* larvae from the different treatment groups. Increased GFP and AR fluorescence in various bony structures are denoted by white and blue arrows, respectively, in BS- and LL-treated fish. One-way ANOVA and Tukey’s multiple comparison tests are used, and statistical significance was set at *p* < 0.05. Different letters denote statistically significant differences between experimental groups. The scale bars, given in the left bottom corner of the first image in the top row, correspond to 100 µm.

**Figure 5 ijms-23-04748-f005:**
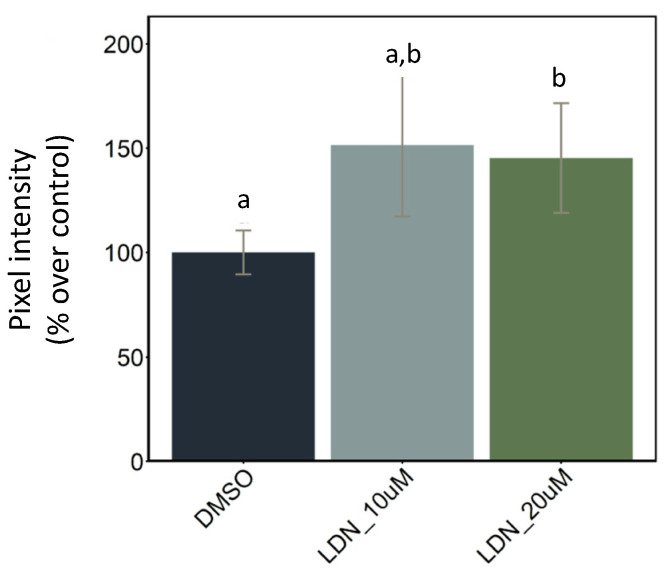
Effects of LDN212854 on *sp7* expression. Integrated pixel intensity values (mCherry red fluorescence) for 5 dpf *Tg(Ola.Sp7:mCherry)* larvae measured in ventral view treated with 10 µM and 20 µM LDN212854 from 2 dpf to 4 dpf. Different letters denote statistically significant differences between experimental groups (one-way ANOVA, *p* < 0.05, followed by Tukey’s post hoc test).

**Figure 6 ijms-23-04748-f006:**
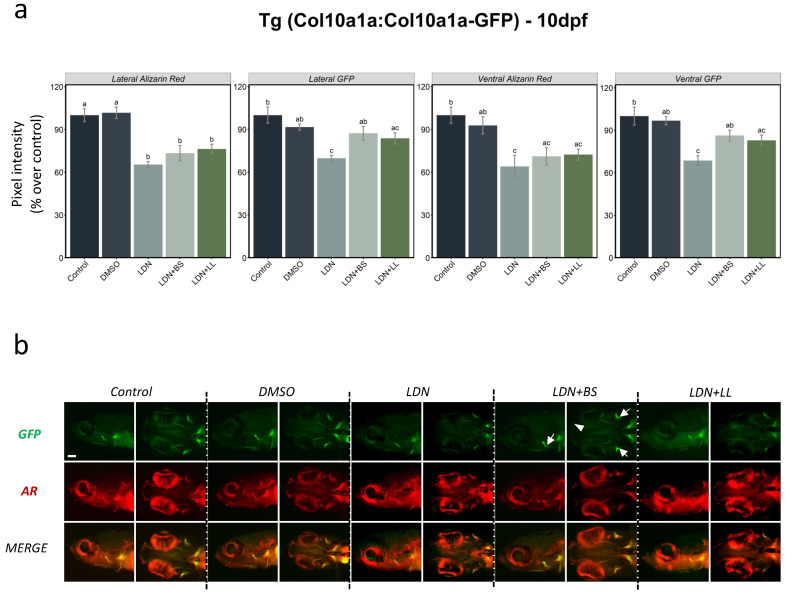
BMP inhibitor exposure followed by two probiotics treatments in *Tg(col10a1a:col10a1a-GFP*) larvae. (**a**) Integrated pixel intensity values for various areas measured in 10 dpf zebrafish *Tg(col10a1a:col10a1a-GFP)* larvae divided into various groups—control, DMSO, LDN, LDN+BS and LDN+LL—and stained with AR. DMSO was used as additional control since it was employed as the solvent for LDN. One-way ANOVA and Tukey’s multiple comparison tests were used, and statistical significance was set at *p* < 0.05. Different letters denote statistically significant differences among the experimental groups. (**b**) GFP, AR staining and merged images of the head area (lateral and ventral views) of *Tg(col10a1a:col10a1a-GFP)* larvae of the different treated groups. White arrows denote GFP in various bony structures and the white arrowhead indicates the presence of a signal in additional structures (that was absent in other groups) in BS-treated fish after LDN exposure (LDN + BS).

**Table 1 ijms-23-04748-t001:** List of primers used for RT-PCR.

Gene Acronym	NCBI Gene Accession No	Forward	Reverse
*col10a1a*	NM_001083827.1	CCCATCCACATCACATCAAA	GCGTGCATTTCTCAGAACAA
*runx2b*	NM_212862.2	GTGGCCACTTACCACAGAGC	TCGGAGAGTCATCCAGCTT
*spp1*	NM_001002308.1	GAGCCTACACAGACCACGCCAACAG	GGTAGCCCAAACTGTCTCCCCG
*cyp26b1*	NM_212666.1	GCTGTCAACCAGAACATTCCC	GGTTCTGATTGGAGTCGAGGC
*sp7*	NM_212863.2	AACCCAAGCCCGTCCCGACA	CCGTACACCTTCCCGCAGCC
*bglap*	NM_001083857.3	GCCTGATGACTGTGTGTCTGAGCG	AGTTCCAGCCCTCTTCTGTCTCAT
*rpl13a*	NM_212784.1	TCTGGAGGACTGTAAGAGGTATGC	AGACGCACAATCTTGAGAGCAG
*rplp0*	NM_131580.2	CTGAACATCTCGCCCTTCTC	TAGCCGATCTGCAGACACAC

## Data Availability

The complete set of images used in this study will be made available in an institutional repository (https://hdl.handle.net/2268/289903; accessed on 24 March 2022).

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
