# Peer review of "Probiotics Enhance Bone Growth and Rescue BMP Inhibition: New Transgenic Zebrafish Lines to Study Bone Health"

_ijms, 2022, doi:10.3390/ijms23094748_

Round 1

Reviewer 1 Report

The study presented by Jerry Maria Sojan and co-workers investigates the effects of gut microbiota or probiotics on bone homeostasis. The authors present two newly generated transgenic zebrafish lines obtained by inserting the coding sequence a fluorescent protein using the CRISPR/Cas9 method into the coding region of the endogenous regulatory regions for the osteoblast marker sp7 gene and the osteoblast- and hypertrophic chondrocyte-specific collagen type X alpha 1a chain which is the downstream target gene of sp7. They describe the expression pattern of these genes and apply them to test the modulatory effects of two probiotics, Bacillus subtilis and Lactococcus lactis, in osteoblast differentiation and early skeletal growth of zebrafish. Additionally, the authors check the possible ability of these probiotics to counteract the deleterious effect of BMP inhibitor treatment on the bone matrix. The results obtained showed that these probiotics modulate some genes related to skeletal development such as runx2, sp7, spp1 and col10a1a and this fact is interpreted as a supporting evidence of their positive effect on bone health. Furthermore, Bacillus subtilis was found to have a more potent effect than Lactococcus lactis, being the former able to significantly reverse the inhibition of bone matrix formation when larvae were exposed to the BMP inhibitor LDN212854.

The paper is interesting since the information provided suggest a new viewpoint in the use of probiotics as a prophylactic treatment in improving bone growth health. The manuscript contains sufficient noteworthy information to justify its publication. I have only one minor point to improve the manuscript:

1. The generation of the transgenic lines is reported in two sections: M&M and Results. I think that the generation should be reported only in the M&M section and describe the characterization of the two new lines in the Results section.

2. The authors provide low-resolution images and the quantification of relative expression levels. It would help to have higher resolution pictures.

3. In the Figures of the relative expression levels, the legends point out that “different letters denote statistically significant differences between experimental groups” but it is not easy to found the correspondence between the different letters and the significant levels.

4. The scale bars are not always presented in the images and when present, they are too small and are barely visible.

5. I understand that it is an additional study but I think it would be very interesting to characterize the histological characteristics of the bones of transgenic lines generated. Previous histological analyses have shown that both cellular and acellular bones can even occur within the same skeletal element in Zebrafish and even two different types of endochondral ossification have been reported. Then, the work would be much more complete with an additional histological analysis.

6. I think that the discussion include a one paragraph on the limitations of the Zebrafish model since it is important to be aware of existing drawbacks to take full advantage of the zebrafish as a model of human diseases. I think it is important to point out that due to the extra whole genome duplication compared to mammals, about 20% of the zebrafish genes have two functional copies, complicating the generation of knock-out disease models. Furthermore, some of the duplicated genes have functionally diverged, thus limiting the use of zebrafish in accurately modeling human diseases.

Reviewer 2 Report

Dear authors, in the following I like to review your study “Probiotics enhance bone growth and rescue BMP inhibition: new transgenic zebrafish lines to study bone health”. The authors have generated 2 new reporter strains of zebrafish and utilized one previously published one to investigate the role of two probiotic bacterial strains on the bone development. These bacterial strains can partially circumvent the chemical inhibition of the BMP-Receptor.

In the following I have some overall questions to the planning of the study:

Why do you use Collagen Type 10 as a reporter for bone? It is true that Coll10 is produced by hypertrophic chondrocytes. But in osteoblasts the expression is reduced. At least for mammals during enchondral ossification, collagen type 10 is removed later on. Thus your collagen reporter and your investigation are not really overlapping on a temporal scale.

Why do your two SP7 reporters show so different patterns? The SP7 tomato and the SP7 GFP should stain exactly the same regions at the same time (Figure 2). And you state the differences, but can you provide an explanation for your finding? This is important to evaluate your tools, because the whole study is dependent on the reliability of the reporters.

I am sorry, do not understand the letters of your statistical test on the bar graphs.

The method of the integrated pixel intensity is not clear to me. Where do you put your cutoff, is it defined by the control? Is it the Alizarin Red Intensity in the GFP positive areas or vice versa? Also these are non-linear dependencies (pixel integration to mineralization /GFP abundance) which makes an intensity comparisons unreliable. I always compare areas where the intensity reaches a defined threshold.

The discussion is more or less a repetition of the results. You should put your data more in the context of the overall research situation.

Finally, would it be necessary to use dead bacteria as control to the living ones to link the role of probiotic metabolism to the bone health?

All together I like the hypothesis and the structure of your study. However as you see, there are several questions to be addressed prior to publication, albeit if they can be addressed properly no additional experiments are required.

Round 2

Reviewer 2 Report

All requests are met to my satisfaction